# Ethnic-Racial (Mis)Match between Mentors and Mentees on Perceived Strength of Relationship

**Jennifer Koide \*, Heather L. McDaniel and Michael D. Lyons** 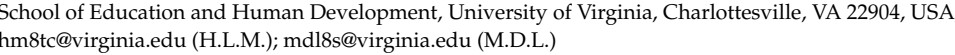

School of Education and Human Development, University of Virginia, Charlottesville, VA 22904, USA; hm8tc@virginia.edu (H.L.M.); mdl8s@virginia.edu (M.D.L.)

\* Correspondence: jk6zx@virginia.edu

**Abstract:** Mentoring programs are popular mechanisms for promoting positive youth development due to developmental research suggesting that youth with strong relationships with a special adult have a lower likelihood of a range of negative outcomes. Community-based mentoring programs are mechanisms for promoting access to mental health support and positive youth outcomes. Youth mentorship programs reveal mixed and modest outcomes, yet youth outcomes are more robust for high-quality relationships. Ethnic-racial identity is hypothesized to affect the relationship quality because shared identities are thought to facilitate trust and empower youth to succeed. However, studies have found that ethnic-racial match does not affect the strength of the dyadic relationship. This study attempts to address these disparate findings by explicitly examining ethnic-racial matching and relationship quality at two-time points. We conducted a two-step hierarchical regression to determine whether ethnic-racial match affected mentee reports of the relationship at follow-up. The results show ethnic-racial match did not significantly predict positive dimensions of the relationship at follow-up. Moreover, same-ethnic-racial matches reported a lower rate of change and slightly less relationship dissatisfaction at time 2 compared to cross-ethnic-racial matches. Results contribute to existing literature showing mixed results in ethnic-racial matching in youth mentorship programs.

**Keywords:** mentorship programs; youth; social-emotional development; mental health; ethnicity; race; racial identity; ethnic-racial match

## 1. Introduction

Relationships between youth and adults across settings, such as those formed in mentoring programs, are important contexts for prevention interventions [1]. Formal, community-based, youth mentoring programs foster strong personal relationships between youth and adults and have widely been used as interventions to reduce risk of psychosocial problems such as depression, substance abuse, and truancy [2]. In addition, previous studies have demonstrated that formal youth mentorship programs are associated with small to medium positive effects on a variety of youth social, academic, and behavioral outcomes [3,4]. Recent meta-analyses of mentoring programs posit the strength of the relationship functions as a mediator to these outcomes [3–5]. The quality of mentoring relationships appears to be a particularly important determinant of both match length and youth outcomes, as high-quality relationships are associated with longer matches and more positive youth outcomes [6]. Moreover, relationship quality may change over the course of the mentoring relationship, which in turn can affect youth outcomes [7]. A strong mentoring relationship has been found to increase youth outcomes such as academic performance, social functioning, and improved friendships [6,8,9].

However, differences in the ethnic-racial backgrounds of mentors and mentees have been thought to affect the development and quality of mentoring relationships. Yet, empirical studies examining the impacts of shared (referred to as same-ethnic-racial match) or dissimilar ethnic-racial (referred to as cross-ethnic-racial match) identities on mentoring

processes show mixed effects. One reason for these effects may be explained by quantitative methodology limitations associated with little variability in match characteristics, small sample sizes, and the use of cross-sectional data [10–12]. This study explicitly examines how ethnic-racial matching affects relationship quality over time using data from a large sample of youth participating in a nationwide mentoring program. By measuring relationship quality and ethnic-racial matching using a longitudinal study design, this study adds to the field's understanding of how demographic characteristics may relate to one important proximal mentoring outcome–relationship quality.

### 1.1. Youth Mentoring Programs

Youth mentoring programs are popular prevention activities focused on adult-youth relationships that operate within youth's ecological contexts. Formal youth mentoring programs pair non-familial adult volunteers with youth to promote healthy development [8,13–15]. Mentoring programs have historically focused on enhancing social relationships, emotional well-being, cognitive skills, and positive identity development [1,4,12,13,16]. From the Ecological Systems Theory perspective, mentoring programs foster relationships between youth and adults which are part of the microsystem ecologies in which youth develop [1,17]. Additionally, some ecological models more explicitly describe how individual, contextual, and developmental factors shape ethnic-racial identity [17–20]. Specific to mentoring, Chandler and colleagues proposed an ecological perspective of mentoring that emphasizes how mentoring outcomes are influenced by individual, dyadic, and social spheres [21]. Thus, the mentoring relationship is shaped by multiple systems, including the ontogenic (e.g., individual traits) system, microsystem (e.g., relationships), and macrosystem (e.g., social norms; power dynamics; Ref. [19]). These various systems include mentor and mentee race, gender, the impact of cross-gender and racial pairings, and social norms and power dynamics. One proposed influence on the quality of mentoring relationship is the ethnic and racial identity match or mismatch between mentors and mentees.

Mentoring programs may prioritize matching mentors and mentees with the same ethnic-racial identity in hopes of better relational and youth outcomes, despite a dearth of research investigating how cultural factors affect youth-adult relationships [8,17,22]. Many mentoring programs prioritize matching children with mentors with the same ethnic-racial background because it is hypothesized that dyads who share an ethnic-racial identity are more likely to have more positive youth outcomes due to stronger feelings of trust. Yet, mentors in formal youth mentoring programs are disproportionately White, middle-class adults, and mentees are often economically disadvantaged and youth of color [8,18,22–25]. Champion and colleagues estimated that 72% of youth in Big Brothers Big Sisters were racial minorities, compared to 32% of mentors, leaving mentoring programs the dilemma of how and whether to match the limited number of ethnic-racial mentors of color with mentees of color [22].

Few studies have examined the impact of ethnic-racial match on mentoring relationship outcomes, and those that have resulted in null or unexpected effects, such as cross-ethnic-racial matches resulting in more positive outcomes than same-ethnic-racial matches [8,24]. On the one hand, matching based on ethnic-racial identities provides mentees with mentors who are more equipped to navigate similar social barriers and stereotypes, which may result in stronger connections and better youth outcomes [24]. Previous research has shown that when mentors are paired with youth of the same racial identity, they are more competent at helping youth navigate social barriers associated with race [14]. On the other hand, research has found that cross-race dyads reported maintaining longer relationships than same-race dyads [12]. Another study found no significant differences between cross-race and same-race dyads in youths' academic performance or social-emotional adjustment [26].

Among studies examining ethnic-racial matching in mentoring, differences in methodological approaches may explain these heterogeneous findings [14,27,28]. Qualitative research on formal mentoring programs has identified that both mentors and mentees

view sharing an ethnic-racial identity as beneficial to the relationship quality [27,28]. Other qualitative studies have found that White mentors describe having difficulty bridging cultural differences such as the role of ethnicity, race, class, and family in mentees' lives [28]. Quantitative studies of ethnic-racial matching on mentoring outcomes have largely used demographic reports of ethnicity and race to assess how ethnic-racial processes influence mentoring outcomes. However, few studies have investigated the role of relationship quality as it relates to ethnic-racial matching, and none have investigated the development of relationship quality in same-and cross-ethnic-racial matches [27,28]. Although quantitative studies of racial-ethnic matching allow researchers to make inferences about mentoring processes across a larger sample, these studies have also been critiqued for limited variability in match characteristics and underpowered statistical analyses due to small sample sizes [28]. Quantitative studies of racial-ethnic matching additionally allow for many racialized mechanisms such as relationship quality and attitudes towards class, ethnicity, and race to be lost. Collectively, the body of work suggests that there are underlying racial constructs in youth mentoring relationships that are not accounted for solely by measuring individual demographics.

One common proximal outcome used to assess the success of a mentoring relationship is the relational quality between mentors and mentees [10–12]. Simply having a mentor does not necessarily yield positive youth outcomes; instead, the relationship quality of the dyad is thought to be a predictor of youth outcomes [10,29–31]. Relationship quality also has the potential to change over the course of the relationship, which can impact youth outcomes—a relationship at baseline may continuously improve, sharply improve, plateau, or decline [7,32]. Relationship quality in the mentoring relationship is influenced by multiple factors such as empathy, mutuality, duration of match, frequency of meetings, and demographics [8,30,33–36]. Previous studies also suggest that mentor support for mentee's ethnic-racial identity, openness to diversity, and training surrounding cultural differences can strengthen relationship quality in cross-ethnic-racial matches [11,14,37].

### 1.2. Current Study

This study examines if ethnic-racial match between mentor and mentee affects the quality of the relationship reported by mentors and mentees over time. Given evidence of mixed effects of matching by ethnic-racial identities on mentoring outcomes, this study seeks to expand upon literature on ethnic-racial matching in youth mentoring programs by including an important determinant of youth outcomes, relationship quality [11,14,37]. Using data from a large nationwide sample of youth participating in a national mentoring program, the study examines whether there are differences in youth perceptions in relationship strength when youth are matched with mentors who share a self-reported ethnic-racial identity as compared to youth who do not share self-reported ethnicity or racial identity with their mentor. We hypothesize that same-ethnic-racial matches will report stronger relationships compared to cross-ethnic-racial matches at baseline and follow-up based on research indicating that dyads with the same ethnic-racial background have shared social experiences, which may lead to strong perceptions of their relationships [8].

## 2. Materials and Methods

### 2.1. Participants and Procedure

Participants were 6636 mentors and mentees from a nationwide youth mentoring program in the United States who completed the Strength of Relationship Survey upon matching at baseline and 9 months (±4 weeks) after baseline to accommodate for matches that end after the traditional school year in the United States. Mentor and mentee dyads were not assigned randomly by the mentoring program and were assigned based on compatibility in personalities and preferences. Participants self-reported data on gender, age, and ethnic-racial identity (Table 1). Mentors identified as 58.4% female compared to 56.1% of mentees. Participants responded to a single self-reported item that asked participants to self-identify as one or more ethnic-racial groups including "American

Indian or Native American", "Asian", "Black", "White", "Other", "Hispanic", or "two or more races". Due to the small sample sizes (n < 100) of mentors and mentees endorsing "American Indian or Native American", "Asian", "two or more races", these groups were excluded from the study and analysis, and not reported. Among mentors and mentees in the analytic sample, 53.2% (n = 3528) of dyads shared an ethnic-racial identity. One dummy-coded indicator called "ethnic-racial match" was created to indicate if mentors and mentees reported a shared ethnic-racial identity (coded as 1), or if they reported differently (coded as 0).

The analytic sample consisted of mentors and mentees participating in a national mentoring program in the United States between 2014 and 2018 who fully completed the baseline and 9-month follow-up surveys. The mentoring agency was a non-profit that served youth aged 5 to 15 years, most of whom came from economically disadvantaged backgrounds. The program matched local youth with local adult mentors, had nationwide locations, and recruited both mentors and mentees from local communities nationwide. Program staff oriented adult volunteers to the program's goals, conducted interviews with potential volunteers, ran background checks on applicants, and requested three letters of recommendation. Mentors were matched with a local mentee and provided them with ongoing social-emotional support for at least 9 months.

*2.2. Measures*

2.2.1. Demographic Characteristics

Mentor and mentee demographics were collected at the beginning of mentor and mentee participation in the program (Table 1). Participants self-reported information related to race, ethnicity, gender, and age.

**Table 1.** Descriptive statistics.

| Demographics | N | Percent |
|---|---|---|
| Mentee Gender | | |
| Female | 3720 | 56.06 |
| Male | 2915 | 43.93 |
| Mentor Gender | | |
| Female | 3873 | 58.36 |
| Male | 2763 | 41.64 |
| Mentee Racial-Ethnic Identity | | |
| Black | 2819 | 42.48 |
| White | 2472 | 37.25 |
| Hispanic | 1345 | 20.27 |
| Mentor Racial-Ethnic Identity | | |
| Black | 859 | 12.94 |
| White | 5232 | 78.84 |
| Hispanic | 545 | 8.21 |
| Racial-Ethnic Match | | |
| No | 3108 | 46.84 |
| Yes | 3528 | 53.16 |

2.2.2. Youth Strength of Relationship Scale

The Youth (Y-SOR) version is a 10-item instrument that assesses mentees' perceptions of positive and negative qualities of the match relationship. Items included in the positive dimension include "My Big has lots of good ideas about how to solve a problem" and "When I am with my Big, I feel safe". Items pertaining to the negative dimension include "When I'm with my Big, I feel mad" and "When I'm with my Big, I feel disappointed". Responses are measured using a five-point Likert-type scale, ranging from 1 (never true) to 5 (always true), scores so that higher values indicate a stronger relationship. As such, higher scores on the positive subscale indicate greater relational satisfaction while higher scores on the negative subscale indicate greater relationship dissatisfaction. The Y-SOR

positive dimension has good internal consistency, $\alpha = 0.76$, as does the negative dimension, $\alpha = 0.68$ [29]. Mentees included in analyses completed the Y-SOR at baseline and at 9-months ($\pm 4$ weeks) post-baseline assessment. Psychometric studies of the Y-SOR have validated the factor structure and have established ethnic-racial invariance of the measure [38].

*2.3. Analytic Approach*

All data analyses were conducted using StataBE 17.0 [39]. Bivariate Pearson correlations assessed the magnitude of the associations between scores for the positive and negative domains at separate time points.

To test whether ethnic-racial match affected mentees' perceptions of positive and negative aspects of strength of relationship, a series of two-step hierarchical multiple regression models were used. Step 1 included mentor and mentee age, gender, mentee ethnic-racial identity and Youth-Strength of Relationship (Y-SOR) positive or negative scores at baseline. Then, ethnic-racial match and the interaction between ethnic-racial match and Y-SOR positive or negative scores at baseline were entered at step 2. The ratio of variance explained by predictors to the total variance of the outcome is reported in addition to incremental variance explained at each step. Standardized betas for both models are reported.

**3. Results**

Results from the correlation matrix revealed different patterns of relationships between Y-SOR subscales (Table 2). Y-SOR positive scores at time 1 and 2 were positively and statistically significantly correlated with each other ($r = 0.29$, $p < 0.01$), as were Y-SOR negative scores ($r = 0.21$, $p < 0.01$. Correlations between the positive and negative subscales were very small but significant and in the expected negative direction ($r$'s $= -0.07$ to $-0.21$; all $p$'s $< 0.01$).

**Table 2.** Correlation table for Y-SOR positive and negative dimensions at time 1 and time 2.

|  | **M** | **SD** | **1** | **2** | **3** | **4** |
|---|---|---|---|---|---|---|
| 1. Y-SOR Positive T1 | 27.95 | 3.42 | - |  |  |  |
| 2. Y-SOR Positive T2 | 28.24 | 3.36 | 0.29 * | - |  |  |
| 3. Y-SOR Negative T1 | 4.26 | 0.96 | −0.14 * | −0.07 * | - |  |
| 4. Y-SOR Negative T2 | 4.34 | 1.10 | −0.09 * | −0.21 * | 0.21 * | - |

Note: M = mean; SD = standard deviation; Youth-Strength of Relationship (Y-SOR); * $p < 0.01$.

*3.1. Positive Dimension of Strength of Relationship*

Results of Step 1 of model testing indicated that mentor and mentee demographic information did not significantly predict reports of positive aspects of the strength of relationship (Table 3). Positive Y-SOR scores at time 2 were predicted by positive Y-SOR scores at time 1 ($\beta = 0.29$, $p < 0.01$). Demographic information and positive SOR scores at time 1 accounted for 9% of the variance in positive Y-SOR scores at time 2. Step 2 included the ethnic-racial match of the dyad, the results indicated that the interaction between ethnic-racial match and positive Y-SOR scores at time 1 did not significantly predict Y-SOR scores at time 2 ($\beta = -0.06$, $p = 0.53$). Even with the addition of ethnic-racial match and the interaction term, the amount of variance explained by the covariates remained at 9%. Moreover, White mentees reported more positive relationships compared to Black mentees ($\beta = 0.04$, $p < 0.05$).

**Table 3.** Coefficients and standard coefficients for two-step hierarchical regression models for positive dimensions of the Y-SOR.

|  | Coefficient | Standardized Beta (ß) | $R^2$ | $\Delta R^2$ |
|---|---|---|---|---|
| **Step 1** |  |  | 0.09 | 0.09 |
| Mentee gender | −0.03 | −0.05 |  |  |
| Mentor gender | 0.05 | 0.04 |  |  |
| Mentee age | 0.00 | 0.00 |  |  |
| Mentor age | 0.00 | 0.00 |  |  |
| Mentee ethnic-racial identity (ref: Black) |  |  |  |  |
| Hispanic | 0.03 | 0.02 |  |  |
| White | 0.02 | 0.02 |  |  |
| Y-SOR time 1 score | 0.29 | 0.29 * |  |  |
| **Step 2** |  |  | 0.09 | 0.09 |
| Mentee gender | −0.03 | −0.05 |  |  |
| Mentor gender | 0.04 | 0.04 |  |  |
| Mentee age | 0.00 | 0.00 |  |  |
| Mentor age | 0.00 | 0.00 |  |  |
| Mentee ethnic-racial identity (ref: Black) |  |  |  |  |
| Hispanic | 0.03 | 0.02 |  |  |
| White | 0.04 | 0.04 * |  |  |
| Y-SOR time 1 score | 0.29 | 0.30 * |  |  |
| Ethnic-racial match | 0.03 | 0.03 |  |  |
| Ethnic-racial match × Y-SOR time 1 score | −0.01 | −0.06 |  |  |

Note: Estimate is significant at the 0.05 level (2-tailed); Youth-Strength of Relationship (Y-SOR); * $p < 0.05$.

### 3.2. Negative Dimension of Strength of Relationship

Step 1 results indicated that mentors' age (ß = 0.03, $p < 0.01$) and negative SOR scores at time 1 (ß = 0.20, $p < 0.01$) predicted negative Y-SOR scores at time 2 (Table 4). Moreover, Hispanic mentees reported less relationship dissatisfaction compared to Black mentees (ß = −0.04, $p < 0.01$). Demographic information and negative Y-SOR scores at time 1 accounted for 4% of the variance in negative Y-SOR scores at time 2. Step 2 results revealed that ethnic-racial match predicted greater negative Y-SOR scores at time 2 (ß = 0.12, $p < 0.01$). However, this effect of ethnic-racial match and the Y-SOR time 1 scores must be considered in the context of the statistically significant interaction. The interaction between ethnic-racial match and Y-SOR scores at time 1 also predicted Y-SOR scores at time 2 (ß = −0.13, $p < 0.01$). This relationship suggested that for youth in mentoring relationships with an ethnic-racial match, more relationship dissatisfaction at baseline was associated with less relationship dissatisfaction at time 2 than for youth in relationships that did not have an ethnic-racial match (Figure 1). Even with the addition of ethnic-racial match, the amount of variance explained by the covariates remained at 4%.

**Table 4.** Coefficients and standard coefficients for two-step hierarchical regression models negative dimensions of the Y-SOR.

|  | Coefficient | Standardized Beta (ß) | $R^2$ | $\Delta R^2$ |
|---|---|---|---|---|
| **Step 1** |  |  | 0.04 | 0.04 |
| Mentee gender | −0.01 | −0.05 |  |  |
| Mentor gender | 0.01 | 0.03 |  |  |
| Mentee age | 0.00 | 0.00 |  |  |
| Mentor age | 0.00 | 0.03 * |  |  |
| Mentee ethnic-racial identity (ref: Black) |  |  |  |  |
| Hispanic | −0.03 | −0.04 * |  |  |
| White | 0.00 | 0.00 |  |  |
| Y-SOR time 1 score | 0.23 | 0.20 * |  |  |

**Table 4.** *Cont.*

|  | Coefficient | Standardized Beta (ß) | R² | ΔR² |
|---|---|---|---|---|
| **Step 2** |  |  | 0.04 | 0.04 |
| Mentee gender | −0.01 | −0.05 |  |  |
| Mentor gender | 0.02 | 0.03 |  |  |
| Mentee age | 0.00 | −0.01 |  |  |
| Mentor age | 0.00 | 0.03 * |  |  |
| Mentee ethnic-racial identity (ref: Black) |  |  |  |  |
| Hispanic | −0.03 | −0.04 * |  |  |
| White | 0.00 | 0.00 |  |  |
| Y-SOR time 1 score | 0.26 | 0.23 * |  |  |
| Ethnic-racial match | 0.06 | 0.12 * |  |  |
| Ethnic-racial match × Y-SOR time 1 score | −0.06 | −0.13 * |  |  |

Note: Correlation is significant at the 0.05 level (2-tailed); Youth-Strength of Relationship (Y-SOR); * $p < 0.05$.

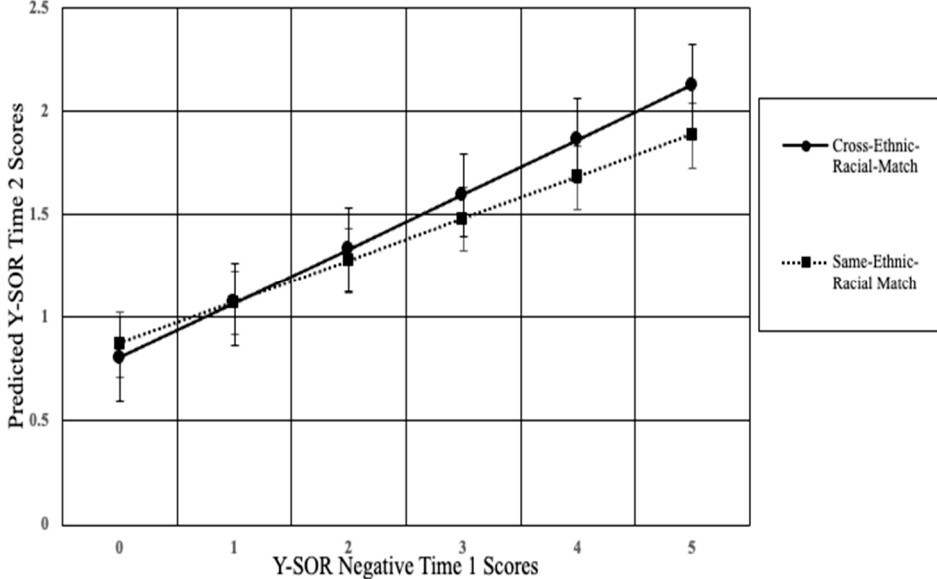

**Figure 1.** Regression coefficients for Y-SOR negative time 1 scores and predicted Y-SOR time 2 scores. Note: Youth-Strength of Relationship (Y-SOR).

## 4. Discussion

Despite the popularity of formal youth mentoring programs, quantitative studies have found a small-to-moderate positive effect on youth academic, behavioral, and social outcomes [3,4]. The quality of relationship between mentors and mentees is hypothesized to impact youth outcomes. Matching by ethnic-racial identity is posited to promote more positive youth outcomes by facilitating a stronger relationship [7,29]. Yet previous quantitative studies show mixed findings despite qualitative studies suggesting that skills such as cultural competence are more impactful on the quality of the relationship compared to matching by demographics [8,14,22]. Qualitative studies have pointed to mentor's low awareness of personal biases and cultural differences as explanations for heterogenous youth outcomes [28].

This study aimed to explicitly examine the role of ethnic-racial matching in the quality of the relationship, as a potential explanation for the heterogenous outcomes in quantitative studies measuring youth outcomes. Across positive and negative subscales of the strength of relationship measure, baseline relationship scores were predictive of follow-up scores. In terms of positive dimensions of the relationship, the results suggest that cross- and same-ethnic-racial matches reported similar scores in relationship quality after 9-months

when controlling for age, gender, and baseline scores. In contrast, results from the negative subscale revealed that matches who shared racial-ethnic identities and higher baseline levels of relationship dissatisfaction reported less relationship dissatisfaction at post-assessment compared to cross-ethnic-racial-matches (ß = −0.06; see Figure 1). Though the effect sizes are small, this suggests that ethnic-racial matching may be protective against additional future relationship dissatisfaction.

This study is one of few to explicitly investigate how ethnic-racial identity of mentors and mentees impacts the relationship quality over time [14,40]. Relationship quality is an important predictor in youth outcomes, and there have been few studies explicitly measuring ethnic-racial matching as a predictor for relationship quality [28,41]. The findings contribute to literature suggesting ethnic-racial matching alone is insufficient in predicting factors, like relationship quality, that can impact youth outcomes. Youth mentoring programs may match dyads based on demographic characteristics such as ethnicity, race, and gender, hypothesizing that mentors and mentees who share common identities may have stronger relationships [7,8,22,29]. However, others have suggested that a shared identity does not necessarily mean they are equipped with the necessary skills to support youth from different cultural backgrounds [18,42]. Qualitative studies, for example, have pointed to specific skills such as ethnocultural empathy, cultural competence, and mentors' comfort with discussing cultural differences as important mechanisms through which mentoring relationships predict relationship quality and subsequent youth outcomes [28,43]. The results of this study, in conjunction with qualitative studies showing the importance of racialized skills such as mentors' cultural competence, self-efficacy have important implications for youth mentoring programs. First, the results of our study contribute to existing literature implying that ethnic-racial matching is not necessarily predictive of strong relationship quality, which is a strong predictor of positive youth outcomes. This study attempted to determine whether ethnic-racial matching was a predictor of relationship quality, yet the results of the study suggest null effects of race-ethnic matching on relationship quality. However, we did find a small significant interaction between race-ethnicity match and baseline levels of negative relationship quality on future perceptions of negative mentoring relationships. This finding suggested that racial matching may have a small protective effect against future negative feelings in a mentoring relationship, when there were higher levels of baseline negative relationship. Overall, we find that matching by ethnicity and/or race had small to null effects on future relationship quality.

The results of the study highlight a gap in research as qualitative studies have found that mentors' skills to bridge cultural gaps are more predictive of a quality relationship than matching by demographics alone. These skills, such as ethnocultural empathy and mentor self-efficacy for racial equity are described in qualitative studies, but often is not measured in quantitative studies. Another takeaway from this study is the need for onboarding and continual training to support mentors with skills required to bridge cultural differences. As evidenced by the results of our study, in addition to previous literature, sharing an ethnic-racial identity is not sufficient in predicting strong quality relationships [29,40]. Previous literature shows that the implementation of programmatic practices like mentor training surrounding bridging cultural gaps and ongoing support throughout the relationship is predictive of stronger relationships and subsequent youth outcomes [35,42].

### 4.1. Limitations

Despite the strengths of the study, there are several limitations and areas for future research. First, the sample size of mentors and mentees who identified as "American Indian or Native American", "Asian", "two or more races", were excluded from the analysis due to a small sample and subsequent low statistical power. It is possible that certain ethnic-racial identities that were not included in the analysis benefit more from having a mentor that shares that identity.

This study also did not examine how identities intersect and whether there are interaction effects between gender and ethnic-racial identity on the strength of the relationship.

Given research suggesting that gender matching in mentoring programs solicit greater youth outcomes and the heterogeneity of outcomes when matching by ethnicity and race [2,3]. Future research can examine how multiple identities interact and whether that impacts the quality of the mentoring relationship.

Last, this study used a quantitative study design to address how ethnic-racial matching relates to relationship quality. This approach is limited in that it does not encapsulate skills referenced in qualitative studies that have been shown to result in stronger mentoring relationships. Future quantitative research should explore how these skills operate in concert with ethnic-racial-match to associated with relationship quality. This study further emphasizes methodological limitations in using demographic information as a proxy for racialized processes such as ethnocultural empathy, mentor self-efficacy surrounding bridging cultural gaps, and other identity related skills that impact relationship quality. Finally, mentors and mentees were not randomized to ethic-racial matched versus unmatched relationships. As such, the relationships discussed are associative and not causal in nature.

### 4.2. Conclusions

The results of this study provide opportunities for change in methodological and measurement approaches in matching mentors and mentees by ethnic and racial identities. The results of this study contribute to a breadth of quantitative research showing null effects of matching by ethnic-racial identity on strength of relationship. However, our results suggest matching mentors and mentees by ethnic-racial identity may be protective longitudinally for matches who initially report a poor relationship quality. Future research should consider how cultural constructs other than ethnic-racial identity contribute to relationship quality between mentors and mentees.

**Author Contributions:** Conceptualization, J.K. and M.D.L.; methodology, J.K.; formal analysis, J.K. and M.D.L.; data curation, M.D.L.; writing—original draft preparation, J.K.; writing—review and editing, J.K., H.L.M. and M.D.L.; supervision, H.L.M. All authors have read and agreed to the published version of the manuscript.

**Funding:** This research received no external funding.

**Institutional Review Board Statement:** Informed consent was obtained from all subjects involved in the study through Institutional Review Board for the Social and Behavioral Sciences at the University of Virginia (approval number: 3243 on 8/19/2020).

**Informed Consent Statement:** Informed consent was obtained from all subjects involved in the study.

**Data Availability Statement:** The datasets presented in this article are not readily available because they were obtained from a third party. Requests to access the data should be directed to Michael Lyons (mdl8s@virginia.edu).

**Conflicts of Interest:** The authors declare no conflict of interest.

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
