# Peer review of "Ethnic-Racial (Mis)Match between Mentors and Mentees on Perceived Strength of Relationship"

_education, doi:10.3390/educsci14040398_

Round 1

Reviewer 1 Report

Comments and Suggestions for Authors

Introduction

Study premise – use of large, national sample. Other studies examining this topic have also used national samples (e.g., Rhodes et al., 2002). It would be helpful to better articulate the study’s rationale.

I think it’s important to specify what type of programs the authors include in their discussion as this has implications. For example, in the discussion of qualitative studies on pg. 3 second paragraph (and later in the Discussion section), some of the studies cited focus on youth-initiated mentoring rather than community-based mentoring. It’s likely that there are differences in the relative importance of same/different race/ethnic match in programs where youth nominate their mentors versus those where they are matched.

Pg. 1, line 28-31: “In addition, youth mentors have been found to enhance youth’s social relationships and emotional states, improve youth cognitive skills, and promote positive identity development (Rhodes, 2006)” – this is a theoretical model. It’s important to clarify that here and/or include studies that have found these outcomes.

Pg. 2, line 47: I’m unclear what the limitations refers to here.

Pg. 2, line 85-86: “Yet, mentors are disproportionately White, middle-class adults, and mentees are often economically disadvantaged and youth of color..” – important to specify that this is the case in formal mentoring programs.

Pg. 3 “Quantitative studies of racial-ethnic match allow for many racialized mechanisms such as attitudes towards class, ethnicity, and race to be lost. Collectively the body of work suggests that there are underlying racial constructs that are not accounted for solely by measuring individual demographics.” – This seems to suggest that this study will do something different to address this, yet I’m not sure that it does.

Methods:

- “American Indian or 163 Native American,” “Asian,” “two or more races” were excluded from analyses due to small sample sizes. How was the group “other” treated in analyses? It seems difficult to assume that the “other” endorsed by mentees and mentors would reflect a shared or different identity? “Other” also not reflected in Table 1.

 Discussion:

-The discussion is unclear. The authors set out to examine ethnic-racial matchups using quantitative methods, yet in the Discussion (e.g., pg. 10, first paragraph) the authors highlight qualitative studies, particularly the nuance that these studies provide. Would their study not be limited in the same way that they highlight? I’m not clear on why this is a point to provide support for/discussion of their findings. Perhaps it’s better suited for the limitations section?

-“ While mentors with shared ethnic-racial identities may be able to relate to their mentors, researchers have suggested that a shared identity does not necessarily mean they are equipped to support youth from different cultural backgrounds (Albright et al., 2017).” This is unclear.

Minor Edits throughout:

-e.g., Pg. 10, line 334: “While mentors with shared ethnic-racial identities may be able to relate to their mentors…” – should read relate to their mentees

Reviewer 2 Report

Comments and Suggestions for Authors

The topic is interesting and relevant. Reference to secondary literary sources is sufficient. The survey covers a small statistical sample, it is not representative. The research methodology is not described in sufficient detail. The arguments and discussion of the findings are not sufficiently coherent, balanced and convincing. To better focus the discussion at the end of the article.

Comments on the Quality of English Language

Minor editing of the English language is required.
